# Research of Curing Time and Temperature-Dependent Strengths and Fire Resistance of Geopolymer Foam Coated on an Aluminum Plate

**Van Su Le** *[ID] **and Petr Louda** [ID]

Department of Material Science, Faculty of Mechanical Engineering, Technical University of Liberec, Studentská 2, 461 17 Liberec, Czech Republic; petr.louda@tul.cz
*   Correspondence: su.le.van@tul.cz or longsuvp90@gmail.com

**Abstract:** Geopolymer foam (GF) uses a potassium activator and can be cured at high temperatures, which can improve its mechanical properties. In this study, we attempted to test this hypothesis by comparing the flexural and compressive strength, apparent density and fire resistance of GF. The composition of the GF used in the experiment included a potassium activator, basalt ground fiber and aluminum powder with a mass ratio to the binder of 0.45, 0.3 and 0.015, respectively. The samples were cured at room temperature and at 50, 70, 85 and 105 °C with a curing time of 2, 4, 6, 12 and 24 h. Then, the samples were kept until being tested on the third, seventh, 14th and 28th day. The results showed that the flexural and compressive strength and apparent density improved and stabilized after seven days at 85 °C. Furthermore, the GF exhibited a substantial increase after three days in its flexural strength by 111% and compressive strength by 122.9% at the optimal temperature of 85 °C for 2 h compared to the values at RT after 28 days. The GF had an apparent density of 0.558–0.623 g/cm$^3$ on the 28th day. As a new alternative to aluminum materials, investigating the fire resistance of sandwich panels (an aluminum plate covered with a GF layer) is important for their safe impregnation. Sandwich panels with thicknesses of 10–20 mm were exposed to a gas fire. The test results showed that the sandwich panels had significantly improved fire resistance compared to unprotected panels. The longest fire resistance times for the aluminum plate coated with 20 mm of GF layer thickness was 7500 s. Thus, the GF coated on the aluminum plate exhibited superior fire resistance and a reduced heat transfer rate compared to uncoated panels.

**Keywords:** strength; aluminum; potassium activator; coating; recycling; foam; fire resistance





## 1. Introduction

Research into geopolymerized materials is a research trend of sustainable models; these are studied to create environmentally friendly production processes, reduce CO$_2$ emissions and utilize industrial wastes such as fly ash, red mud, furnace slag, etc., to create products with high usability. The wide geopolymer applications make chemical anti-corrosive products useful for fast-curing applications in the high-tech field and in medicine; this is especially true for geopolymer foam (GF) as a fire-resistant material. This study is part of a series of studies from the Technical University of Liberec (TUL) on geopolymer composites. The sources of raw materials such as basalt fiber, furnace slag and sand to produce GF have become rich and varied. However, the selection of the materials involved in forming geopolymers is still based on the priority criteria of minimizing industrial waste and increasing environmental sustainability. The curing process of the geopolymer composite has a significant impact on the quality and durability of the material, particularly as a result of curing temperature and time. The recommended curing temperature of geopolymer composite is in the temperature range of room temperature (RT) to 135 °C [1–9]. Pavel Rovnaník [10] reported that the compressive strength of metakaolin-based geopolymer was reduced and its pore size increased as the sample curing temperature was increased to

80 °C. Bai et al. [9] showed a reduction in the compressive strength of metakaolin-based geopolymer foams with high total porosity. Hamad [11] indicated that the small size of specimens gave a higher compressive strength to high-performance lightweight foamed concrete compared to other sizes. GF has been widely considered for application in the field of fire-resistant materials due to its special properties such as low thermal conductivity, high temperature resistance and very light weight. Therefore, research into the application of GF for fire-retardant purposes is important. Studies on GF have used it as a surface coating to protect substrate materials such as wood, concrete and metal [12–14] Davidovits et al. [12] reported on a panel manufacturing process in which a panel coated with a layer of an alkaline silicate mixture was cured a temperature of at least 80 °C. Sakkas et al. [13] showed that a concrete slab coated with a 50 mm thick K-geopolymer layer significantly improved the fire-resistance of the material as a thermal barrier. Temuujin et al. [14] indicated that the adhesion of the metakaolin-based geopolymer coatings strongly depended on the composition of the material coated on metal substrates as thermal barriers.

GF is a lightweight material with a density of 200 to 1000 kg/m$^3$ [1,2,15], stability at high temperatures [16,17] and fire-retardance in the case of the geopolymer having a potassium activator [13,18,19], being quickly installed at low-temperatures [20–23] and having thermal insulation [24–28]. GFs are considered as building materials [29–35], membranes and membrane supports [36–38], adsorbents and fillers [39–42] and catalysts [17,43]. The processing method of GFs is the thermal expansion of K-nano-poly(siloxo) at temperatures above 250 °C. In addition, GFs can be produced by the geochemical method using foaming agents such as hydrogen peroxide (H$_2$O$_2$) [44,45], aluminum powder [23,26,35], sodium perborate [31] and silica fume [46,47]. GFs can develop into a low-density material with thermal insulation properties using thermal expansion agents or chemical methods [48].

Stone wool fiber is a base-building material with insulation properties that is used for buildings and furnace applications. It has good properties such as being light-weight, insulated and suitable for application to temperatures below 700 °C; furthermore, it has a high melting point 1500 °C and is an environmentally friendly material. In addition to its application in insulation, it improves the mechanical strength of concrete. Light-weight fiber-reinforced geopolymer can be used up to the high temperature of 1000 °C [49]. Fiber-reinforced GF has been shown to have good mechanical and heat-resistant properties [10]. The output of this fiber by the Rockwool Company is about 300 kilotons per year; as the manufacturer declares, the recycling rate of this yarn is 100%. To increase the recycled applications for this fiber, we made GF-reinforced milled fiber to create a cheap geopolymer product with good enough properties for high-temperature resistance.

There is very little research on the optimum curing conditions for geopolymers. In particular, the studies into the optimal curing conditions of GF are based on the evaluation of its durability criteria. Studies on the optimal curing conditions of GF are very important as they help to reduce the time and cost of the production process. Besides, to the authors' knowledge, there has been no research into fibers used for fire-resistant purposes in aluminum sheets covered with protective geopolymer layer-reinforced ground fiber.

This work presents the mechanical properties of GF that were developed in different curing temperatures and time conditions. The optimum condition for the curing of GF was synthesized and investigated in terms of its compressive and flexural strength and apparent density. The composition of the GF used in the experiment included potassium activator, basalt ground fiber and aluminum powder with a mass ratio to the binder of 0.45, 0.3 and 0.015, respectively. The samples were cured at room temperature and 50, 70, 85 and 105 °C with a curing time of 2, 4, 6, 12 and 24 h. Then, the samples were kept until they were tested on the third, seventh, 14th and 28th day. The fire resistance tests analyzed the fire resistance of the aluminum plate coated by the GF layer. The composition of the studied GF was taken from previous research, with GF types prepared at room temperature as the mixture, has excellent mechanical properties at the age of 28 days.

## 2. Materials and Methods

### 2.1. Materials

Baucis LK-type cement is commercially distributed by the company České Lupkové Závody a.s, Nove Straseci, Czech Republic, and was used as the inorganic material. Aluminum powder (pkchemie Inc., Trebic, Czech Republic) was used to create porosity in the GF structure. It had a chemical composition of 99.4% Al, 0.16% Fe, 0.06% Si and 0.001% Cu by weight, and the average grain size D50 was 51.47 μm. Ground fiber was used in the experimental work as a reinforcement material. To obtain ground fibers, a stone wool slab (manufactured by Saint-Gobain Construction Product, CZ a.s., Sokolovska, Czech Republic) was milled. It had a density of 150 kg/m$^3$, thermal conductivity 0.037 W/(mK) and range of temperature operation of up to 700 °C. Chemical element compositions of raw materials are described in Table 1 and were measured by a scanning electron microscope—ZEISS Ultra Plus—fitted with an energy-dispersive spectroscope (EDS) and operated at 15 kV (ZEISS, Berlin, Germany).

**Table 1.** Chemical element compositions of raw materials.

| Constituents | O | Si | Ca | Al | K | Mg | Ti | Na | Fe | Mn | S | Other | LOI |
|---|---|---|---|---|---|---|---|---|---|---|---|---|---|
| Geopolymer | 46.9 | 20.8 | 12.6 | 15.3 | 0.62 | 1.34 | 0.78 | 0.18 | 0.57 | 0.22 | 0.18 | 0.51 | 2.56 |
| Basalt ground fiber | 39.4 | 31.8 | 18.0 | 9.2 | 0.6 | 0.4 | 0.3 | 0.3 | 0.1 | – | – | – | 2.05 |

LOI: Loss on Ignition.

### 2.2. Sample Preparation

The samples in this study were made using five parts by weight of cement and four parts by weight of activator, which was recommended by the suppliers of Baucis LK, and the mixture was stirred for five minutes at RT until the solution was homogenized. Next, the geopolymer was mixed with 30% basalt ground fiber by comparing the ratio of cement, and the mixture was homogenized for a further five minutes. Aluminum powder was then added at the end of the mixture preparation for about one minute at high speed. Directly after mixing, the fresh GFs were cast into molds. Plastic sheets then covered the samples until they became hard enough to be removed.

### 2.3. Characterizations

The samples were tested at different curing temperatures and different aging times. The purpose of this was to analyze the conditions for which the GF must be prepared in order to have best mechanical properties. The specimens were cured for 2, 4, 6 12 and 24 h at room temperature and 50, 70, 85 and 105 °C. For each temperature, three-point flexure, compressive test and apparent density measurements were set after 3, 7, 14 and 28 days.

The apparent density of GF was measured according to standard ČSN EN 1936. The mass, height, width and length of each sample were measured to calculate the apparent density of the sample measurement for the flexure test.

The strength tests used a universal testing machine—Instron (Model 4202, Labortech s.r.o, Opava, Czech Republic). The sample dimensions for the flexure test were 40 mm by 40 mm by 160 mm (Figure 1a); for each test, three samples were tested. The flexural tests were conducted with a crosshead speed of 2.0 mm/min at RT (about 22 ± 3 °C) and a span length of 120 mm. For the compression test, the broken parts from the samples used in the flexure test were used, making a total of six samples for this case (Figure 1b); their dimensions were 40 mm × 40 mm × 40 mm (Figure 1c).

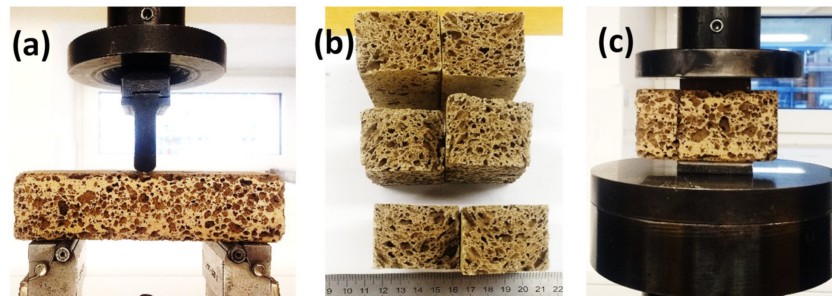

**Figure 1.** (**a**) Three-point flexure test, (**b**) broken samples, (**c**) compression test.

The fire resistance tests were conducted on aluminum plates coated with a GF layer. Fire resistance specimens were fabricated by coating the GF mixture on the aluminum plates' surface by casting, and plates were cured for 28 days at RT before testing. The thicknesses of the aluminum plates before coating with the GF layer were 2 mm. The fire resistance samples had a 2D dimension of 300 mm × 300 mm. The fire exposure region of the samples was 200 mm × 200 mm. The thicknesses of the coated GF layers for the fire resistance specimens were measured from three samples with different thicknesses of 10, 20 mm and uncoated. The inside and outside temperatures of the fire test furnace were measured using in-built thermocouples connected to the computer. The thermocouples' hot sides and cold sides were mounted on the fire-exposed surface of the specimen and the unexposed sample surface, respectively. A system of natural gas heated the test furnace. The endpoint of the fire resistance test for aluminum specimens was reached when the measured temperature on the unexposed surface of samples reached 210 °C using an infrared thermometer—C. A. 1950 Diaca (CHAUVIN ARNOUX GROUP, Asnières-Sur-Seine, France). The furnace fire was controlled to obtain the heating rate recommended in the standard ISO 834.

## 3. Results and Discussion

Figures 2 and 3 show the effect of curing temperature and time on the flexural and compressive strength of the geopolymer at 3, 7, 14 and 28 days. GF cured at RT reached a flexural strength of 1.27 MPa and compressive strength of 2.75 MPa at 28 days. The flexural, compressive strength and apparent density of GF cured at ambient temperature after 28 days gave a reference result for comparing investigated values. The strength of GF with an early curing age and cured at 50 °C was not significantly lower than those of the GFs cured at 70, 85 and 105 °C due to the slow settling of the geopolymer mixture. A higher temperature increases the hard structure formation of the GF, thereby increasing its strength. The flexural and compressive strengths of the GF cured at 50, 70, 85 and 105 °C, respectively, reached their final values 2 h after mixing. The flexural and compressive strengths after being cured for three days at 85 °C for 2 h increased 111% and 122.9%, respectively, compared to the samples cured at RT after 28 days. After 24 h at 85 °C, the flexural and compressive strengths after three days increased, respectively, 119.7% and 128.4% compared to those at RT for 28 days. Meanwhile, GF cured at 50 °C grew slowly, and its quality was better than at the start of the experiment. However, the compressive strength after 28 days was not significantly lower than that of the GF cured at RT. In contrast, although the mixtures cured at 70, 85 and 105 °C exhibited rapid strength development, they achieved their target strength values three days after mixing. In the early ages, the compressive strength increased with increasing temperature; the level of geopolymerization was higher, increasing the samples' strength. On the other hand, with longer ages, when the geopolymerization level was approximately the same, the samples' strengths were similar. The flexural strengths of test pieces cured at different temperatures and times showed a similar development trend to compressive strength. The flexural and compressive strengths of GF were described as a function of the curing time at different temperatures. The final values of compressive strength were reached after about three days,

whereas these values were reached for samples cured at RT after 28 days. The curing times of the strength development of the GFs treated at 50, 70, 85 and 105 °C were very similar. Samples cured for 2 h at high temperature reached their final strength after three days with flexural and compressive strength values of about 2.8 and 1.2 MPa, respectively. As observed in Figure 3c, the compressive strengths of GF after 3 days at 85 °C after soaking of 2, 4, 6, 12 and 24 h, respectively, were 3.38, 3.36, 3.15, 3.32 and 3.53 MPa, and they increased by 22.9%, 22.2%, 14.5%, 20.7% and 28.4% compared to the sample cured at RT after 28 days. Meanwhile, when the GF was cured at 85 °C, the development trend of the GF compressive strength was very similar at 3, 7, 14 and 28 days. In this case, the compressive strength at three days was equal to or higher than 3.2 MPa, but furthermore, the final compressive strength was not reduced.

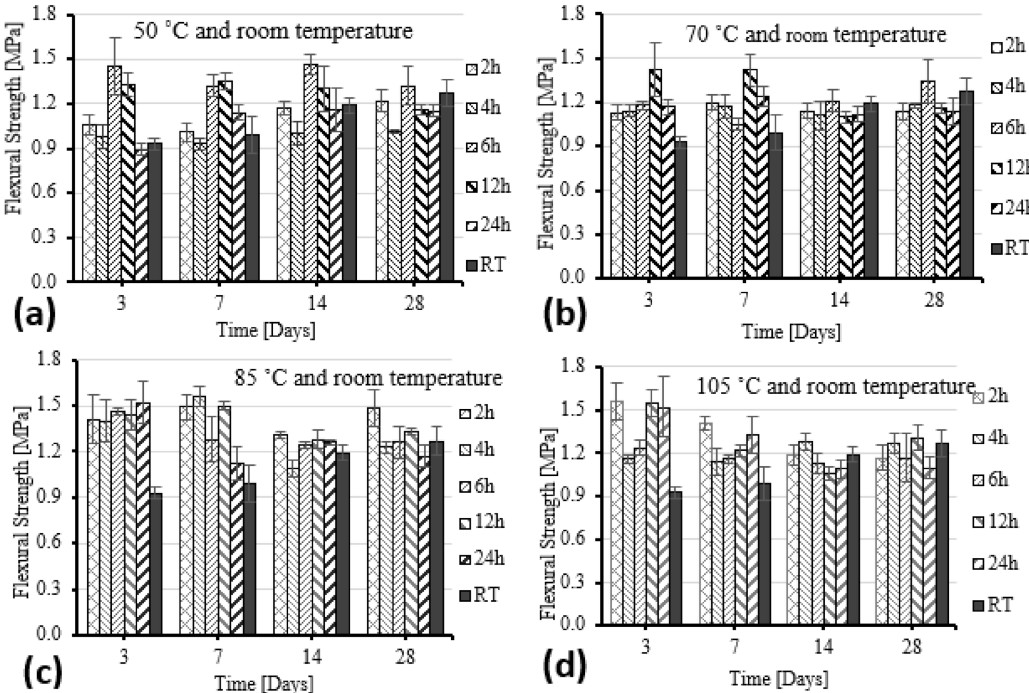

**Figure 2.** Flexural strengths of geopolymer foam (GF) at different curing temperatures: (**a**) 50 °C, (**b**) 70 °C, (**c**) 85 °C and (**d**) 105 °C.

Interestingly, the compressive and flexural strength of GF samples were most significantly improved under curing conditions of 85 °C, whereas the samples cured at 105 °C showed a decrease in their compressive and flexural strength. This reduction was due to the rapid curing and the fast water loss from the geopolymer structure [8,45]. Besides, all samples showed significant improvements in compressive and flexural strength as curing conditions increased from 50 to 105 °C compared to the values at RT. The improvement in the mechanical strength of GFs under curing at 85 °C could be expressed in terms of curing and porosity [10].

The apparent density of GF at different curing temperatures is depicted in Figure 4. The apparent density of GF cured at RT decreased over time and ranged from 0.706 to 0.586 g/cm$^3$. It was shown that the increase in curing time and temperature slightly significantly decreased the apparent density of GF. Furthermore, the sample becoming lighter over time may have been due to the extra loss of unreacted water during geopolymerization. In studies, it has been shown that the density of GF after 28 days at room temperature is stable [8,19,50–52]. In all results regarding density at the age of 3 days, there was a higher apparent density value after curing for 28 days. At seven days, the apparent density of all samples decreased compared to its apparent density at three days. Then, it became almost constant after seven days, and the apparent density of specimens cured at 50, 70, 85 and

105 °C was significantly unchanged, as visible in the red line for 0.586 g/cm$^3$ in Figure 4. The results indicated that the apparent density of GF after seven days at all temperature checks was equivalent to its apparent density cured at 28 days. Therefore, the apparent density values of the GF were stable on the seventh day. The condition and mechanical properties of the GF samples from this work and the literature are depicted in Table 2, in which the results of the investigation are presented regarding GF treated at 40, 60, 70 and 80 °C for 24, 96 and 168 h and they were measured after seven or more days. The results of the study were compared with previous studies with time correlation.

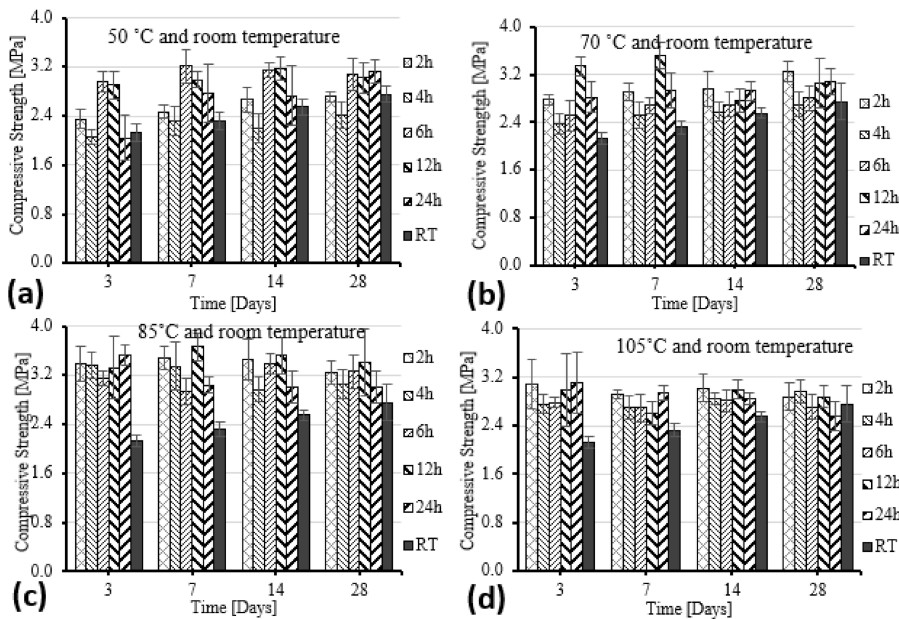

**Figure 3.** Compressive strength of GF at different curing temperatures: (**a**) 50 °C, (**b**) 70 °C, (**c**) 85 °C and (**d**) 105 °C.

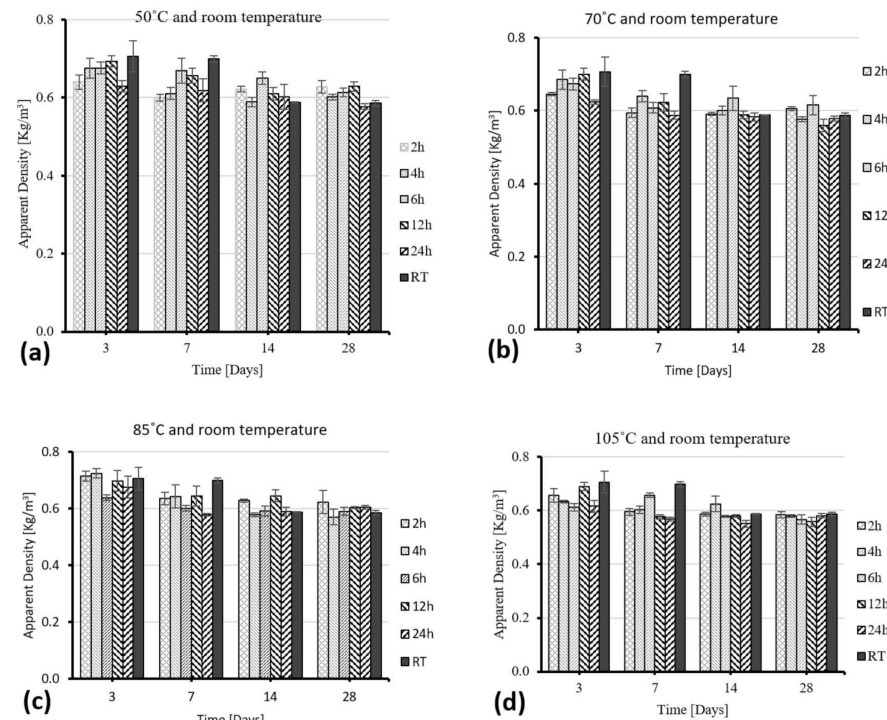

**Figure 4.** Apparent density of GF at different curing temperatures: (**a**) 50 °C, (**b**) 70 °C, (**c**) 85 °C and (**d**) 105 °C.

**Table 2.** The condition and mechanical properties of GF from this work and literature.

| CTI (h) | CTE (°C) | TT (Days) | CS (MPa) | FS (MPa) | D (g/cm³) | Ref. |
|---|---|---|---|---|---|---|
| 5 | 80 | 28 | 0.3–3 | – | 0.45–0.75 | [8] |
| 10 | 80 | 28 | 0.55 | – | 0.37 | [53] |
| 24 | 60 | 7 | 0.42–1.59 | – | – | [54] |
| 24 | 60 | 28 | 1.45 | – | 0.31 | [55] |
| 24 | 60 | – | 0.57–5.9 | – | 0.21–1 | [5] |
| 24 | 70 | 28 | 3.07 | – | 0.92 | [6] |
| 24 | 40 | 28 | 4.23–14.8 | – | 0.10–0.16 | |
| 24 | 70 | – | 2.9–9.3 | 2–3.6 | 0.64–1 | [7] |
| 24 | 40/75 | 28 | 2.3–30.7 | – | 0.37–0.87 | [16] |
| 24 | 75 | – | 2.19–3.11 | – | 0.4–0.51 | [9] |
| 24 | 85 | – | 0.67–0.96/ | – | 0.239–0.335 | [46] |
| 96 | 40 | 7 | 4.5 | – | 0.54 | [37] |
| 168 | 60 | – | 0.06–1.56 | – | 0.21–0.63 | [4] |
| – | RT | 28 | 1.3 | – | 0.6 | [56] |
| 2/4/6/12/24 | 20/50/70/85/105 | 3/7/14/28 | 2.75–3.5 | 0.9–1.5 | 0.56–0.62 | This work |

CTI: curing time, CTE: curing temperature, TT: test time, CS: compressive strength, FS: flexural strength, D: density, RT: room temperature.

Besides, the increase in the flexural strength of GF was clearly explained by the flexural load–displacement curve analysis. The effects of curing time and temperature and aging on the flexural load–displacement curves of the GF samples are clearly shown in Figure 5, where the samples cured at RT are representative. With soak times increasing at high curing temperatures, the load-bearing capacity can be obviously seen to decrease in Figure 5a. The shape of curves changed positively with an increasing soak time at 85 °C. While samples cured at RT broke smoothly due to the native of the first crack, increasing soak times at high curing temperatures affected the test pieces, the method of breaking of which improved and became a soft fracture.

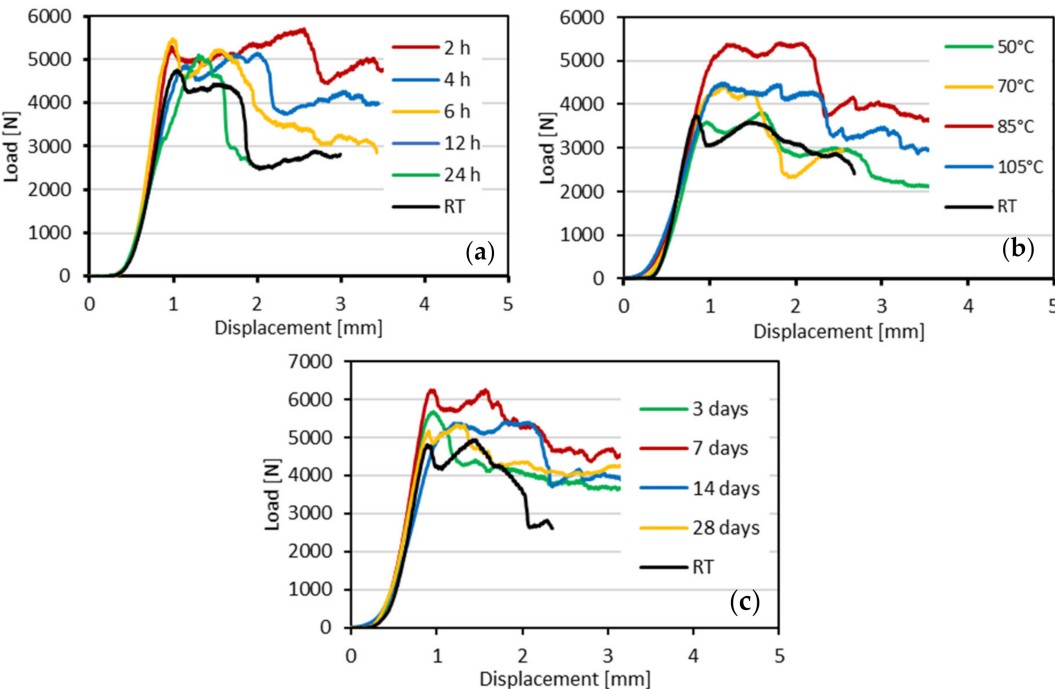

**Figure 5.** Flexural load–displacement behavior of GF under different conditions: (**a**) at 85 °C for 2, 4, 6, 12, 24 and RT after 28 days, (**b**) at 50, 70, 85 and 105 °C and RT for 2 h after 7 days, (**c**) at 85 °C for 2 h after 3, 7, 14 and 28 days.

Furthermore, the dropping length of the flexural load–displacement curves is also precisely related to the rate of reinforcement, and the shapes of the curves are considerably

different. The maximum load for specimens was performed under curing at 85 °C and soaking for 2 h. A similar argument can be put forward for the samples maintained at RT, 50, 70, 85 and 105 °C, as shown in Figure 5b. On the other hand, the loads of the samples with increasing curing time to achieve a hard displacement state were even greater than that at RT, and the maximum load of GF was under curing at 85 °C for 2 h. Figure 5c shows the flexural load–displacement curves of the samples after aging for 3, 7, 14 and 28 days and curing at 85 °C for 2 h. After only 3 days, the sample's load was greater than that at room temperature. At 7 days of age, the load of the sample was maximum. Surprisingly, the load of the test samples decreased after 14 and 28 days.

The failure modes of GFs specimens are described in Figure 6. The effect of compressive force deformed the sample, as shown in Figure 6b, with cracked shapes appearing around the center of the cube, although the sides of the cube were not completely broken. This shows that stress was maintained and that the load was transferred to another region of the matrix through the fibers (Figure 6c). Therefore, the first cracks were not localized with increasing applied load. Instead, new cracks developed elsewhere in the geopolymer matrix. As a result, an increase in the applied load resulted in multiple cracks and a higher load capacity. Besides, the strength of GFs changed with different sizes of the specimen. For instance, the compressive strength of the 50 mm cube increased by 15% compared with the 100 mm cube [11], or the compressive strength determined by cubic specimens was almost 15% higher than that obtained in cylindrical specimens [11,57]. In this work, the specimens tested for compressive strength were cubes with sizes of 40 mm × 40 mm × 40 mm.

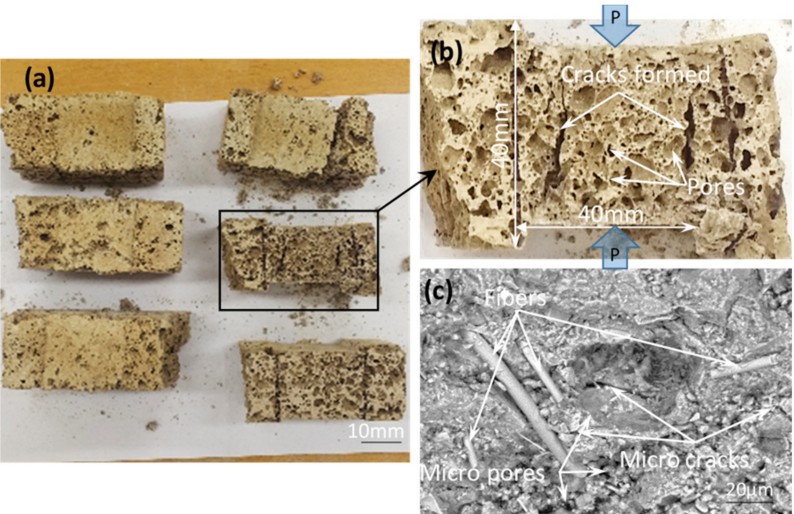

**Figure 6.** Failure modes of GF specimens: (**a**) the compression test of the broken parts, (**b**) surface failure of samples, and (**c**) micrograph of GFs.

Fire resistance testing was performed on the aluminum plates coated by GF layers of different thicknesses using the flame from a gas burner. Temperature–time curves of fire resistance tests for different coating layer thicknesses are shown in Figure 7: (a) uncoated sample, (b) sample with a 10 mm coating and (c) sample with a 20 mm coating. The fire resistance time of the uncoated sample was 250 s. The fire resistance times of GF samples coated with 10 and 20 mm layers were 4000 and 7500 s, respectively. The aluminum plate covered with a protective GF layer of 20 mm in thickness resisted the fire for the longest time. Its fire-resistance time was 30 times longer than that of the aluminum plate without the GF layer. The maximal temperature in the furnace did not exceed 600 °C. The aluminum plate covered with a protective GF layer of 10 mm in thickness showed a fire-resistance time that was 16 times longer than that of the uncoated sample, while the fire temperature in the furnace increased to 600 °C. The top-end condition was determined as a cold-side temperature greater than 210 °C, as measured by an infrared thermometer (C. A. 1950 Diaca (Figure 7d)).

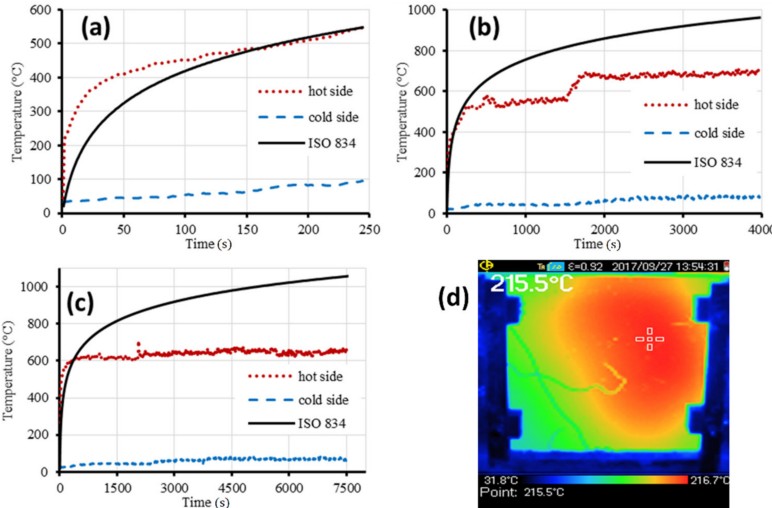

**Figure 7.** Temperature–time curves of fire resistance tests for different coating layer thicknesses: (**a**) uncoated, (**b**) 10 mm, (**c**) 20 mm and (**d**) the cold-side temperature.

Micrographs and photographs of the surface of the GF are shown in Figure 8. Figure 8a,b shows the micropore, fiber and GF matrix in the GF structure at room temperature and after firing. Figure 8b shows that the fibers still exist in the GF structure and there is no change in comparison with the GF structure before and after firing. Figure 8c shows the almost totally distributed porosity of the surface sample. Porosity occupies most of the GF structure, with amorphous pores, which is the cause of the increase in heat transfer in the geopolymer structure [9,22]. The exposed surface of the sample did not show any change in deformation or fracture. The change was visible by the light color. The sample exposed to fire (Figure 8e) had a lighter gray color than the unexposed sample (Figure 8d). The GF exposed to high temperatures had a hardened surface. Temperatures at around 600 °C did not affect the GF structure but made it more stiffened. In addition, it is known that the geopolymer can withstand temperatures as high as 1000 °C [14,19,58–61]. The results of the experiment showed that the aluminum plate coated with a protective GF layer significantly increased its fire resistance time. Therefore, the aluminum plate coated with the GF layer could be used for good insulation and fire prevention at temperatures around 600 °C. Moreover, the geopolymer composite was coated on base substrates such as wood, concrete and steel for passive fire protection [12–14,18,19]. It is clear from this study that the GF is suitable for coating on aluminum plates for fire protection.

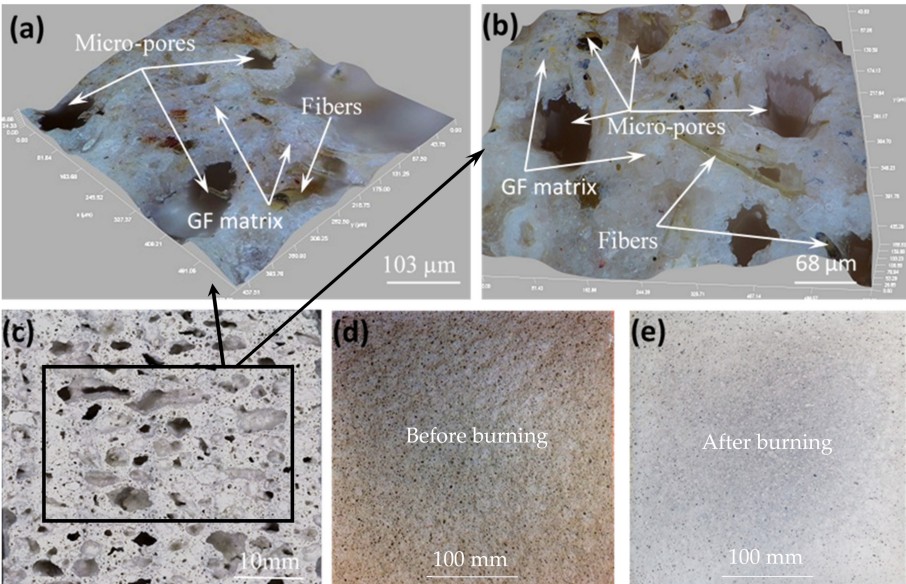

**Figure 8.** Micrographs and photographs of the GF surface: (**a**,**d**) before test, (**b**,**e**) after test and (**c**) cross-section of 40 mm by 40 mm.

## 4. Conclusions

In this research, the effects of curing temperature and time at different temperatures on the mechanical properties of GF with basalt fiber reinforcement and a foaming agent with aluminum powder were investigated by measuring the development of its flexural and compressive strength over time. In addition, this study presented the fire resistance of the studied GF-coated aluminum plate. The experimental and analytical results allow us to make the following conclusions.

- All samples showed significant improvements in compressive and flexural strength as curing conditions increased from 50 to 105 °C compared to at RT. The compressive and flexural strength of GF was most significantly improved under the curing condition of 85 °C.
- The maximum load for specimens was found after curing for 7 days at 85 °C and soaking for 2 h; surprisingly, the load of the test samples decreased after 14 and 28 days.
- The apparent density of the GF was stable at the age of 7 days.
- The aluminum plate covered with a protective GF layer showed an increased fire resistance time compared to the unprotected plate. The aluminum plate covered with a protective GF layer of 20 mm in thickness resisted the fire for the longest time. Its fire-resistance time was 30 times higher than that of the aluminum plate without the GF layer. The maximal temperature in the furnace did not exceed 600 °C.
- Temperatures at around 600 °C did not affect the GF structure but made it more stiffened.

The results indicated that GF has the potential for fast production while retaining its mechanical properties, thereby reducing the demands of the production process, and it is a light-weight material for fire prevention and resistance as a building material. Further studies are necessary in future research papers because the sample's parameters may change after 3, 12, 36 and 60 months.

**Author Contributions:** Conceptualization, methodology, software, validation, formal analysis, investigation, resources, data curation, V.S.L.; writing—original draft preparation, V.S.L.; writing—review and editing, V.S.L. and P.L.; supervision, P.L.; project administration, P.L. All authors have read and agreed to the published version of the manuscript.

**Funding:** This research received no external funding.

**Institutional Review Board Statement:** Not applicable.

**Informed Consent Statement:** Not applicable.

**Data Availability Statement:** The data presented in this study are available in article.

**Acknowledgments:** The publication is grateful to the Ministry of Education, Youth and Sports of the Czech Republic for the financial support of the Institutional Endowment for Long-Term Conceptual Development and the Department of Material Science, Faculty of Mechanical Engineering, Technical University of Liberec.

**Conflicts of Interest:** The authors declare no conflict of interest.

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
