# Peer review of "Research of Curing Time and Temperature-Dependent Strengths and Fire Resistance of Geopolymer Foam Coated on an Aluminum Plate"

_coatings, doi:10.3390/coatings11010087_

Round 1
Reviewer 1 Report
1. Abstract should give clear information
No information of fibres
Material
No fibres mensioned in this part
2. Rivise Part 3 (Result and discussion)
unclear information regarding the results
line 102,122, 123, 128, 129, 150
No discussion of fibre inluence
3. Part 4 (Conclusion)
very short, unclear conclusions regarding aluminium plates
Author Response
Dear reviewer,
Thank you very much for your valuable comments. I have put efforts to respond to your comments in a sufficient way to make the manuscript publishable in Coatings.
Yours sincerely,
Le Van Su
Ref.: Response to the reviewers’ comments on the manuscript ID COATINGS-1027514
Reviewer #1: Comments and Suggestions for Authors:
1. Abstract should give clear information (No information of fibres, Material, No fibres mensioned in this part)
The authors revised the manuscript following the suggestion of the reviewer (see lines 47-70 in the revised manuscript).
2. Rivise Part 3 (Result and discussion) (unclear information regarding the results, line 102,122, 123, 128, 129, 150, No discussion of fibre influence)
The text was revised following the correction of the reviewer (see in the part Result and discussion in the revised manuscript).
3. Part 4 (Conclusion) - very short, unclear conclusions regarding aluminium plates
The conclusion was corrected following the suggestion of the reviewer (see lines 262-278 in the revised manuscript).
Yours sincerely,
Van Su Le
Corresponding author
Reviewer 2 Report
Manuscript No.: coatings-1027514
Title: Effect of curing temperature, time on mechanical properties, and fire resistance to geopolymer foam, which is coated on aluminum plate
Review: This paper investigated the effect of curing temperature and curing time on the fire resistance to geopolymer foam. It is an interesting topic. The authors are encouraged to add more information, such as materials characterization and bonding failure pattern. On the basis of these observations, I recommend a minor revision for this paper. The following remarks are noted during the review.
Comments:
1.The current title is confusing. The english should be improved.
2. Scales are missing in Fig. 6 (d), (e). Are there fibres in Fig. 6(c)?
3. More references are needed. such as the work from Prof. Jianhe Xie.
4. Line 190, 'It is above and below 0.6 g/cm3' ?? confusing.
5. Materials characterization should be added.
6. The bonding failure pattern could be provided.
Author Response
Dear reviewer,
Thank you very much for your valuable comments. I have put efforts to respond to your comments in a sufficient way to make the manuscript publishable in Coatings.
Yours sincerely,
Le Van Su
Ref.: Response to the reviewers’ comments on the manuscript ID COATINGS-1027514
Reviewer #2: This paper investigated the effect of curing temperature and curing time on the fire resistance to geopolymer foam. It is an interesting topic. The authors are encouraged to add more information, such as materials characterization and bonding failure pattern. On the basis of these observations, I recommend a minor revision for this paper. The following remarks are noted during the review. Here are some comments:
1. The current title is confusing. The english should be improved.
The Title manuscript was revised to the introduction section following the suggestion of the reviewer (see lines 2-5).
2. Scales are missing in Fig. 6 (d), (e). Are there fibres in Fig. 6(c)?
Figure 6 were changed following the suggestion of the reviewer (see in lines 253-256).
3. More references are needed. such as the work from Prof. Jianhe Xie.
The manuscript authors thank the reviewer for the suggestion. There have been many articles related to geopolymer materials. However, we only referred to the papers closely related to the research topic (Research of curing time- and temperature-dependent strengths and fire resistance of geopolymer foam coated on aluminum plate). Moreover, authors added several articles to the reference following the suggestion of the reviewer (see in the revised manuscript).
4. Line 190, 'It is above and below 0.6 g/cm3' ?? confusing.
The text is corrected in the part conclusion of the manuscript.
5. Materials characterization should be added.
The materials characterization was added in lines 87-93 following the suggestion of the reviewer.
6. The bonding failure pattern could be provided.
The bonding failure pattern was added in lines 198-221 following the suggestion of the reviewer.
Yours sincerely,
Van Su Le
Corresponding author
Reviewer 3 Report
1. The paper has no scientific grounds. It constitutes simply a report of usual laboratory tests on small sized geopolymerized materials. It does not have innovative points. The paper cannot be accepted in the present form.
2. In Introduction, no overview of the conducted experimental research was provided.
3. It is extremely imported to highlight the novelty of the provided research and experiments. Also, it should be stated what is scientific contribution of the paper, which is not highlighted in this Introduction.
The paper must include full stress-strain curves during compression and load-deflection curves during bending (or load-displacement curves during splitting).
4. The failure modes (crack patterns) of concrete specimens (and their evolution) must be described in detail with sketches.
5. The meaning of the experimental results for the engineering practice must be deeply discussed.
6. The size effect for concrete should be also discussed.
7. In order to make comparison of provided results it is suggested to add existed results from similar research from other researchers and to provide discussion of the obtained results. The discussion of the results should be added with critically explanations of advantages and disadvantages in comparison with similar materials and the obtained results from other researchers.
8. Most of conclusions are trivial. More detailed conclusion regarding obtained result should be provided.
Author Response
Dear reviewer,
Thank you very much for your valuable comments. I have put efforts to respond to your comments in a sufficient way to make the manuscript publishable in Coatings.
Yours sincerely,
Le Van Su
Ref.: Response to the reviewers’ comments on the manuscript ID COATINGS-1027514
Reviewer #3: Comments and Suggestions for Authors
1. The paper has no scientific grounds. It constitutes simply a report of usual laboratory tests on small sized geopolymerized materials. It does not have innovative points. The paper cannot be accepted in the present form.
The manuscript authors thank the suggestion of the reviewer. However, the manuscript has focused on Research of curing time- and temperature-dependent strengths and fire resistance of geopolymer foam coated on aluminum plate
2. In Introduction, no overview of the conducted experimental research was provided.
The overview of the conducted experimental research was revised in the Introduction part.
3. It is extremely imported to highlight the novelty of the provided research and experiments. Also, it should be stated what is scientific contribution of the paper, which is not highlighted in this Introduction.
The novelty of the presented research was revised following the suggestion of the reviewer (see in the Introduction part in the revised manuscript).
3. The paper must include full stress-strain curves during compression and load-deflection curves during bending (or load-displacement curves during splitting).
4. The failure modes (crack patterns) of concrete specimens (and their evolution) must be described in detail with sketches.
The failure modes were described in the revised manuscript (see in Figure 6 and in lines 233-241).
5. The meaning of the experimental results for the engineering practice must be deeply discussed.
Many thanks for the reviewer comment. The discussion part was revised following the comment of the reviewer (see lines 133-256 in the revised manuscript).
6. The size effect for concrete should be also discussed.
The manuscript authors thank the reviewer for the suggestion. we only referred to the papers closely related to the research topic (Research of curing time- and temperature-dependent strengths and fire resistance of geopolymer foam coated on aluminum plate). Furthermore, the authors conducted the test samples according to the standards into the reference at the suggestion of the reviewer. (see in the revised manuscript).
7. In order to make comparison of provided results it is suggested to add existed results from similar research from other researchers and to provide discussion of the obtained results. The discussion of the results should be added with critically explanations of advantages and disadvantages in comparison with similar materials and the obtained results from other researchers.
The discussion of the results was added in the results and discussion part in the revised manuscript.
8. Most of conclusions are trivial. More detailed conclusion regarding obtained result should be provided.
The conclusion was rewritten.
Yours sincerely,
Van Su Le
Corresponding author
Reviewer 4 Report
Dear Authors, Thank You for Your work. Below, I am sending a few comments. Best Regards, Reviewer The Authors wrote: Coatings improve and increase strength after 7 days, but what happens to the sample after 90 days? Has it been checked? The introduction and the analyzes and tests performed deal with the subject of sustainable development and construction, which is the right aspect of the research. Components such as fly ash have been replaced. These are components with an amorphous, heterogeneous structure and are not resistant to temperature or pressure (they are referred to as thermodynamically metastable materials). A material in an amorphous state is a solid, but the molecules that compose it are arranged in a chaotic manner, more similar to that found in liquids. In building materials, depending on the method of their formation, there are, among others, hydrated calcium silicates (C-S-H, C-S-H (I), C-S-H (II)), where especially the C-S-H phase is susceptible to crystallization under appropriate external conditions. Moreover, the amorphous phase very rarely occurs "by itself" in the entire volume of the test substance and materials found in practice, ie material / mineral, but usually coexists with the crystalline phase. The composition of the materials was tested, which is a good basis and an initial test for further tests. Your conclusion: What does it mean that: "Apparent density cured for seven days at 70, 85 and 105 ° C was stabilized." It happens that materials modified with aluminum compounds or treated with aluminum swell. How was the density stabilized? Can the Authors explain this? The research of physical and mechanical properties is interesting, but the article lacks more information on the internal structure of the material under study and information on the processes characteristic of this group of materials (especially since the materials are subject to hydrothermal treatment). Do the Authors consider additional analyzes beyond those presented in the article? I suggest that the Authors continue to study the presented materials, because in 90 days, 12, 24, 36 months the parameters of the samples may change, despite the satisfactory results at present.
Author Response
Dear reviewer,
Thank you very much for your valuable comments. I have put efforts to respond to your comments in a sufficient way to make the manuscript publishable in Coatings.
Yours sincerely,
Le Van Su
Ref.: Response to the reviewers’ comments on the manuscript ID COATINGS-1027514
Reviewer #4: Comments and Suggestions for Authors
Dear Authors, Thank You for Your work. Below, I am sending a few comments. Best Regards, Reviewer The Authors wrote: Coatings improve and increase strength after 7 days, but what happens to the sample after 90 days? Has it been checked? The introduction and the analyzes and tests performed deal with the subject of sustainable development and construction, which is the right aspect of the research. Components such as fly ash have been replaced. These are components with an amorphous, heterogeneous structure and are not resistant to temperature or pressure (they are referred to as thermodynamically metastable materials). A material in an amorphous state is a solid, but the molecules that compose it are arranged in a chaotic manner, more similar to that found in liquids. In building materials, depending on the method of their formation, there are, among others, hydrated calcium silicates (C-S-H, C-S-H (I), C-S-H (II)), where especially the C-S-H phase is susceptible to crystallization under appropriate external conditions. Moreover, the amorphous phase very rarely occurs "by itself" in the entire volume of the test substance and materials found in practice, ie material / mineral, but usually coexists with the crystalline phase. The composition of the materials was tested, which is a good basis and an initial test for further tests. Your conclusion: What does it mean that: "Apparent density cured for seven days at 70, 85 and 105 ° C was stabilized." It happens that materials modified with aluminum compounds or treated with aluminum swell. How was the density stabilized? Can the Authors explain this? The research of physical and mechanical properties is interesting, but the article lacks more information on the internal structure of the material under study and information on the processes characteristic of this group of materials (especially since the materials are subject to hydrothermal treatment). Do the Authors consider additional analyzes beyond those presented in the article? I suggest that the Authors continue to study the presented materials, because in 90 days, 12, 24, 36 months the parameters of the samples may change, despite the satisfactory results at present.
The comment of the reviewer is right. The manuscript was revised and rewritten in some parts following the correction of the reviewer.
Yours sincerely,
Van Su Le
Corresponding author
Round 2
Reviewer 1 Report
line 85/86
blank between number and % should be correct (see line 141)
Author Response
Dear reviewer
Thank you very much for your valuable comments. The comment of the reviewer is right. The text was corrected following the suggestion of the reviewer in the revised manuscript.
Yours sincerely,
Van Su Le
Corresponding author
Reviewer 3 Report
The authors didn't improve or correct all the given comments.
When asked to make a state of the art, it was meant to write what was done in the articles and what conclusions were drawn. not just listing articles:
[1, 2, 14, 15],[16, 17], [21-24], [25-29], [30-36], [37-40], [41-44] [17, 45], [32, 46-48], [24, 27,36], [32], [49, 50].
In order to make comparison of provided results it is suggested to add existed results from similar research from other researchers and to provide comparison and discussion of the obtained results. The discussion of the results should be added with critically explanations of advantages and disadvantages in comparison with similar materials and the obtained results from other researchers.
The authors didn't present similar results and compare them with their.
Author Response
Dear reviewer
Thank you very much for your valuable comments. I have put efforts to respond to your comments in a sufficient way to make the manuscript publishable in Coatings.
Yours sincerely,
Le Van Su